# Nonparametric Multi-group Membership Model for Dynamic Networks

**Myunghwan Kim**
Stanford University
Stanford, CA 94305
mykim@stanford.edu

**Jure Leskovec**
Stanford University
Stanford, CA 94305
jure@cs.stanford.edu

Relational data—like graphs, networks, and matrices—is often dynamic, where the relational structure evolves over time. A fundamental problem in the analysis of time-varying network data is to extract a summary of the common structure and the dynamics of the underlying relations between the entities. Here we build on the intuition that changes in the network structure are driven by dynamics at the level of groups of nodes. We propose a nonparametric multi-group membership model for dynamic networks. Our model contains three main components: We model the birth and death of individual groups with respect to the dynamics of the network structure via a distance dependent Indian Buffet Process. We capture the evolution of individual node group memberships via a Factorial Hidden Markov model. And, we explain the dynamics of the network structure by explicitly modeling the connectivity structure of groups. We demonstrate our model's capability of identifying the dynamics of latent groups in a number of different types of network data. Experimental results show that our model provides improved predictive performance over existing dynamic network models on future network forecasting and missing link prediction.

## 1  Introduction

Statistical analysis of social networks and other relational data is becoming an increasingly important problem as the scope and availability of network data increases. Network data—such as the friendships in a social network—is often dynamic in a sense that relations between entities rise and decay over time. A fundamental problem in the analysis of such dynamic network data is to extract a summary of the common structure and the dynamics of the underlying relations between entities.

Accurate models of structure and dynamics of network data have many applications. They allow us to predict missing relationships [20, 21, 23], recommend potential new relations [2], identify clusters and groups of nodes [1, 29], forecast future links [4, 9, 11, 24], and even predict group growth and longevity [15].

Here we present a new approach to modeling network dynamics by considering time-evolving interactions between groups of nodes as well as the arrival and departure dynamics of individual nodes to these groups. We develop a dynamic network model, Dynamic Multi-group Membership Graph Model, that identifies the birth and death of individual groups as well as the dynamics of node joining and leaving groups in order to explain changes in the underlying network linking structure. Our nonparametric model considers an infinite number of latent groups, where each node can belong to multiple groups simultaneously. We capture the evolution of individual node group memberships via a Factorial Hidden Markov model. However, in contrast to recent works on dynamic network modeling [4, 5, 11, 12, 14], we explicitly model the birth and death dynamics of individual groups by using a distance-dependent Indian Buffet Process [7]. Under our model only *active/alive* groups influence relationships in a network at a given time. Further innovation of our approach is that we not only model relations between the members of the same group but also account for links between members and non-members. By explicitly modeling group lifespan and group connectivity structure we achieve greater modeling flexibility, which leads to improved performance on link prediction and network forecasting tasks as well as to increased interpretability of obtained results.

The rest of the paper is organized as follows: Section 2 provides the background and Section 3 presents our generative model and motivates its parametrization. We discuss related work in Section 4 and present model inference procedure in Section 5. Last, in Section 6 we provide experimental results as well as analysis of the social network from the movie, *The Lord of the Rings*.

## 2 Models of Dynamic Networks

First, we describe general components of modern dynamic network models [4, 5, 11, 14]. In the next section we will then describe our own model and point out the differences to the previous work.

Dynamic networks are generally conceptualized as discrete time series of graphs on a fixed set of nodes $N$. Dynamic network $Y$ is represented as a time series of adjacency matrices $Y^{(t)}$ for each time $t = 1, 2, \cdots, T$. In this work, we limit our focus to unweighted directed as well as undirected networks. So, each $Y^{(t)}$ is a $N \times N$ binary matrix where $Y_{ij}^{(t)} = 1$ if a link from node $i$ to $j$ exists at time $t$ and $Y_{ij}^{(t)} = 0$ otherwise.

Each node $i$ of the network is associated with a number of latent binary features that govern the interaction dynamics with other nodes of the network. We denote the binary value of feature $k$ of node $i$ at time $t$ by $z_{ik}^{(t)} \in \{0, 1\}$. Such *latent features* can be viewed as assigning nodes to multiple overlapping, latent clusters or groups [1, 21]. In our work, we interpret these latent features as memberships to latent groups such as social communities of people with the same interests or hobbies. We allow each node to belong to multiple groups simultaneously. We model each node-group membership using a separate Bernoulli random variable [17, 22, 29]. This is in contrast to *mixed-membership* models where the distribution over individual node's group memberships is modeled using a multinomial distribution [1, 5, 12]. The advantage of our *multiple-membership* approach is as follows. Mixed-membership models (*i.e.*, multinomial distribution over group memberships) essentially assume that by *increasing* the amount of node's membership to some group $k$, the same node's membership to some other group $k'$ has to *decrease* (due to the condition that the probabilities normalize to 1). On the other hand, multiple-membership models do not suffer from this assumption and allow nodes to truely belong to multiple groups. Furthermore, we consider a nonparametric model of groups which does not restrict the number of latent groups ahead of time. Hence, our model adaptively learns the appropriate number of latent groups for a given network at a given time.

In dynamic network models, one also specifies a process by which nodes dynamically join and leave groups. We assume that each node $i$ can join or leave a given group $k$ according to a Markov model. However, since each node can join multiple groups independently, we naturally consider *factorial hidden Markov models* (FHMM) [8], where latent group membership of each node independently evolves over time. To be concrete, each membership $z_{ik}^{(t)}$ evolves through a 2-by-2 Markov transition probability matrix $Q_k^{(t)}$ where each entry $Q_k^{(t)}[r, s]$ corresponds to $P(z_{ik}^{(t)} = s | z_{ik}^{(t-1)} = r)$, where $r, s \in \{0 = \text{non-member}, 1 = \text{member}\}$.

Now, given node group memberships $z_{ik}^{(t)}$ at time $t$ one also needs to specify the process of link generation. Links of the network realize according to a *link function* $f(\cdot)$. A link from node $i$ to node $j$ at time $t$ occurs with probability determined by the link function $f(z_{i\cdot}^{(t)}, z_{j\cdot}^{(t)})$. In our model, we develop a link function that not only accounts for links between group members but also models links between the members and non-members of a given group.

## 3 Dynamic Multi-group Membership Graph Model

Next we shall describe our Dynamic Multi-group Membership Graph Model (DMMG) and point out the differences with the previous work. In our model, we pay close attention to the three processes governing network dynamics: (1) birth and death dynamics of individual groups, (2) evolution of memberships of nodes to groups, and (3) the structure of network interactions between group members as well as non-members. We now proceed by describing each of them in turn.

**Model of active groups.** Links of the network are influenced not only by nodes changing memberships to groups but also by the birth and death of groups themselves. New groups can be born and old ones can die. However, without explicitly modeling group birth and death there exists ambiguity

between group membership change and the birth/death of groups. For example, consider two disjoint groups $k$ and $l$ such that their lifetimes and members do not overlap. In other words, group $l$ is born after group $k$ dies out. However, if group birth and death dynamics is not explicitly modeled, then the model could interpret that the two groups correspond to a single latent group where all the members of $k$ leave the group before the members of $l$ join the group. To resolve this ambiguity we devise an explicit model of birth/death dynamics of groups by introducing a notion of *active* groups.

Under our model, a group can be in one of two states: it can be either active (alive) or inactive (not yet born or dead). However, once a group becomes inactive, it can never be active again. That is, once a group dies, it can never be alive again. To ensure coherence of group's state over time, we build on the idea of distance-dependent Indian Buffet Processes (dd-IBP) [7]. The IBP is named after a metaphorical process that gives rise to a probability distribution, where customers enter an Indian Buffet restaurant and sample some subset of an infinitely long sequence of dishes. In the context of networks, nodes usually correspond to 'customers' and latent features/groups correspond to 'dishes'. However, we apply dd-IBP in a different way. We regard each time step $t$ as a 'customer' that samples a set of active groups $\mathcal{K}_t$. So, at the first time step $t = 1$, we have $Poisson(\lambda)$ number of groups that are initially active, *i.e.*, $|\mathcal{K}_1| \sim Poisson(\lambda)$. To account for death of groups we then consider that each active group at time $t - 1$ can become inactive at the next time step $t$ with probability $\gamma$. On the other hand, $Poisson(\gamma\lambda)$ new groups are also born at time $t$. Thus, at each time currently active groups can die, while new ones can also be born. The hyperparameter $\gamma$ controls for how often new groups are born and how often old ones die. For instance, there will be almost no newborn or dead groups if $\gamma \approx 1$, while there would be no temporal group coherence and practically all the groups would die between consecutive time steps if $\gamma = 0$.

Figure 1(a) gives an example of the above process. Black circles indicate active groups and white circles denote inactive (not yet born or dead) groups. Groups 1 and 3 exist at $t = 1$ and Group 2 is born at $t = 2$. At $t = 3$, Group 3 dies but Group 4 is born. Without our group activity model, Group 3 could have been reused with a completely new set of members and Group 4 would have never been born. Our model can distinguish these two disjoint groups.

Formally, we denote the number of active groups at time $t$ by $K_t = |\mathcal{K}_t|$. We also denote the state (active/inactive) of group $k$ at time $t$ by $W_k^{(t)} = \mathbf{1}\{k \in \mathcal{K}_t\}$. For convenience, we also define a set of newly active groups at time $t$ be $\mathcal{K}_t^+ = \{k | W_k^{(t)} = 1, W_k^{(t')} = 0 \; \forall t' < t\}$ and $K_t^+ = |\mathcal{K}_t^+|$.

Putting it all together we can now fully describe the process of group birth/death as follows:

$$
K_t^+ \sim \begin{cases} Poisson\,(\lambda), & \text{for } t = 1 \\ Poisson\,(\gamma\lambda), & \text{for } t > 1 \end{cases}
$$

$$
W_k^{(t)} \sim \begin{cases} Bernoulli(1 - \gamma) & \text{if } W_k^{(t-1)} = 1 \\ 1, & \text{if } \sum_{t'=1}^{t-1} K_{t'}^+ < k \le \sum_{t'=1}^{t} K_{t'}^+ \\ 0, & \text{otherwise}. \end{cases} \tag{1}
$$

Note that under this model an infinite number of active groups can exist. This means our model automatically determines the right number of active groups and each node can belong to many groups simultaneously. We now proceed by describing the model of node group membership dynamics.

**Dynamics of node group memberships.** We capture the dynamics of nodes joining and leaving groups by assuming that latent node group memberships form a Markov chain. In this framework, node memberships to active groups evolve through time according to Markov dynamics:

$$
P(z_{ik}^{(t)} | z_{ik}^{(t-1)}) = Q_k = \begin{pmatrix} 1 - a_k & a_k \\ b_k & 1 - b_k \end{pmatrix},
$$

where matrix $Q_k[r, s]$ denotes a Markov transition from state $r$ to state $s$, which can be a fixed parameter, group specific, or otherwise domain dependent as long as it defines a Markov transition matrix. Thus, the transition of node's $i$ membership to active group $k$ can be defined as follows:

$$
a_k, b_k \sim Beta(\alpha, \beta), \; z_{ik}^{(t)} \sim W_k^{(t)} \cdot Bernoulli\left( a_k^{1 - z_{ik}^{(t-1)}} \left(1 - b_k\right)^{z_{ik}^{(t-1)}} \right). \tag{2}
$$

Typically, $\beta > \alpha$, which ensures that group's memberships are not too volatile over time.

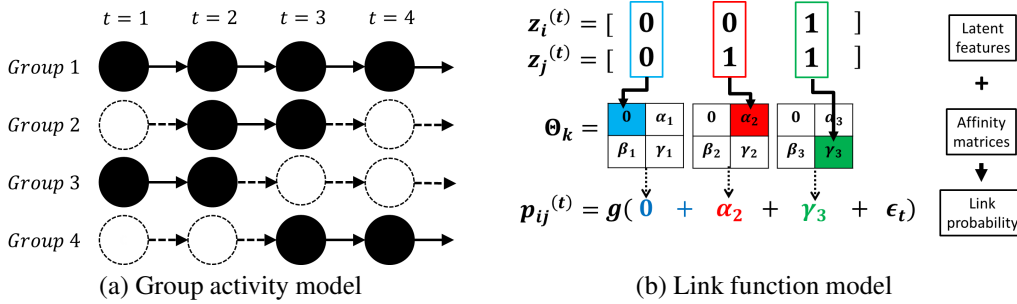

(a) Group activity model    (b) Link function model

Figure 1: **(a) Birth and death of groups:** Black circles represent active and white circles represent inactive (unborn or dead) groups. A dead group can never become active again. **(b) Link function:** $z_i^{(t)}$ denotes binary node group memberships. Entries of link affinity matrix $\Theta_k$ denotes linking parameters between all 4 combinations of members ($z_i^{(t)} = 1$) and non-members ($z_i^{(t)} = 0$). To obtain link probability $p_{ij}^{(t)}$, individual affinities $\Theta_k[z_j^{(t)}, z_j^{(t)}]$ are combined using a logistic function $g(\cdot)$

.

**Relationship between node group memberships and links of the network.** Last, we describe the part of the model that establishes the connection between node's memberships to groups and the links of the network. We achieve this by defining a link function $f(i, j)$, which for given a pair of nodes $i, j$ determines their interaction probability $p_{ij}^{(t)}$ based on their group memberships.

We build on the Multiplicative Attribute Graph model [16, 18], where each group $k$ is associated with a link affinity matrix $\Theta_k \in \mathcal{R}^{2 \times 2}$. Each of the four entries of the link affinity matrix captures the tendency of linking between group's members, members and non-members, as well as non-members themselves. While traditionally link affinities were considered to be probabilities, we relax this assumption by allowing affinities to be arbitrary real numbers and then combine them through a logistic function to obtain a final link probability.

The model is illustrated in Figure 1(b). Given group memberships $z_{ik}^{(t)}$ and $z_{jk}^{(t)}$ of nodes $i$ and $j$ at time $t$ the binary indicators "select" an entry $\Theta_k[z_{ik}^{(t)}, z_{jk}^{(t)}]$ of matrix $\Theta_k$. This way linking tendency from node $i$ to node $j$ is reflected based on their membership to group $k$. We then determine the overall link probability $p_{ij}^{(t)}$ by combining the link affinities via a logistic function $g(\cdot)$[1]. Thus,

$$p_{ij}^{(t)} = f(z_{i\cdot}^{(t)}, z_{j\cdot}^{(t)}) = g\left(\epsilon_t + \sum_{k=1}^{\infty} \Theta_k[z_{ik}^{(t)}, z_{jk}^{(t)}]\right), \quad Y_{ij} \sim Bernoulli(p_{ij}^{(t)}) \qquad (3)$$

where $\epsilon_t$ is a density parameter that reflects the varying link density of network over time.

Note that due to potentially infinite number of groups the sum of an infinite number of link affinities may not be tractable. To resolve this, we notice that for a given $\Theta_k$ subtracting $\Theta_k[0, 0]$ from all its entries and then adding this value to $\epsilon_t$ does not change the overall linking probability $p_{ij}^{(t)}$. Thus, we can set $\Theta_k[0, 0] = 0$ and then only a finite number of affinities selected by $z_{ik}^{(t)}$ have to be considered. For all other entries of $\Theta_k$ we use $\mathcal{N}(0, \nu^2)$ as a prior distribution.

To sum up, Figure 2 illustrates the three components of the DMMG in a plate notation. Group's state $W_k^{(t)}$ is determined by the dd-IBP process and each node-group membership $z_{ik}^{(t)}$ is defined as the FHMM over active groups. Then, the link between nodes $i$ and $j$ is determined based on the groups they belong to and the corresponding group link affinity matrices $\Theta$.

## 4    Related Work

Classically, non-Bayesian approaches such as *exponential random graph* models [10, 27] have been used to study dynamic networks. On the other hand, in the Bayesian approaches to dynamic network analysis *latent variable models* have been most widely used. These approaches differ by the structure of the latent space that they assume. For example, euclidean space models [13, 24] place nodes

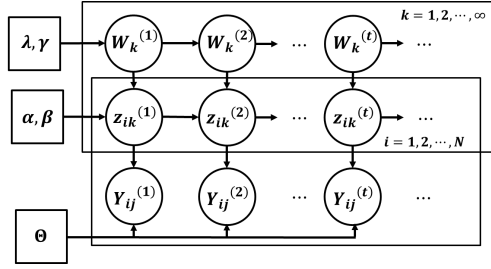

Figure 2: Dynamic Multi-group Membership Graph Model. Network $Y$ depends on each node's group memberships $Z$ and active groups $W$. Links of $Y$ appear via link affinities $\Theta$.

in a low dimensional Euclidean space and the network evolution is then modeled as a regression problem of node's future latent location. In contrast, our model uses HMMs, where latent variables stochastically depend on the state at the previous time step. Related to our work are *dynamic mixed-membership* models where a node is probabilistically allocated to a set of latent features. Examples of this model include the dynamic mixed-membership block model [5, 12] and the dynamic infinite relational model [14]. However, the critical difference here is that our model uses multi-memberships where node's membership to one group does not limit its membership to other groups. Probably most related to our work here are DRIFT [4] and LFP [11] models. Both of these models consider Markov switching of latent multi-group memberships over time. DRIFT uses the infinite factorial HMM [6], while LFP adds "social propagation" to the Markov processes so that network links of each node at a given time directly influence group memberships of the corresponding node at the next time. Compared to these models, we uniquely incorporate the model of group birth and death and present a novel and powerful linking function.

## 5   Model Inference via MCMC

We develop a Markov chain Monte Carlo (MCMC) procedure to approximate samples from the posterior distribution of the latent variables in our model. More specifically, there are five types of variables that we need to sample: node group memberships $Z = \{z_{ik}^{(t)}\}$, group states $W = \{W_k^{(t)}\}$, group membership transitions $Q = \{Q_k\}$, link affinities $\Theta = \{\Theta_k\}$, and density parameters $\epsilon = \{\epsilon_t\}$. By sampling each type of variables while fixing all the others, we end up with many samples representing the posterior distribution $P(Z, W, Q, \Theta, \epsilon | Y, \lambda, \gamma, \alpha, \beta)$. We shall now explain a sampling strategy for each varible type.

**Sampling node group memberships** $Z$**.** To sample node group membership $z_{ik}^{(t)}$, we use the forward-backward recursion algorithm [26]. The algorithm first defines a deterministic forward pass which runs down the chain starting at time one, and at each time point $t$ collects information from the data and parameters up to time $t$ in a dynamic programming cache. A stochastic backward pass starts at time $T$ and samples each $z_{ik}^{(t)}$ in backwards order using the information collected during the forward pass. In our case, we only need to sample $z_{ik}^{(T_k^B : T_k^D)}$ where $T_k^B$ and $T_k^D$ indicate the birth time and the death time of group $k$. Due to space constraints, we discuss further details in the extended version of the paper [19].

**Sampling group states** $W$**.** To update active groups, we use the Metropolis-Hastings algorithm with the following proposal distribution $P(W \rightarrow W')$: We add a new group, remove an existing group, or update the life time of an active group with the same probability $1/3$. When adding a new group $k'$ we select the birth and death time of the group at random such that $1 \leq T_{k'}^B \leq T_{k'}^D \leq T$. For removing groups we randomly pick one of existing groups $k''$ and remove it by setting $W_{k''}^{(t)} = 0$ for all $t$. Finally, to update the birth and death time of an existing group, we select an existing group and propose new birth and death time of the group at random. Once new state vector $W'$ is proposed we accept it with probability

$$\min \left( 1, \frac{P(Y|W')P(W'|\lambda, \gamma)P(W' \rightarrow W)}{P(Y|W)P(W|\lambda, \gamma)P(W \rightarrow W')} \right) . \tag{4}$$

We compute $P(W|\lambda, \gamma)$ and $P(W' \rightarrow W)$ in a closed form, while we approximate the posterior $P(Y|W)$ by sampling $L$ Gibbs samples while keeping $W$ fixed.

**Sampling group membership transition matrix** $Q$. Beta distribution is a conjugate prior of Bernoulli distribution and thus we can sample each $a_k$ and $b_k$ in $Q_k$ directly from the posterior distribution: $a_k \sim Beta(\alpha + N_{01,k}, \beta + N_{00,k})$ and $b_k \sim Beta(\alpha + N_{10,k}, \beta + N_{11,k})$, where $N_{rs,k}$ is the number of nodes that transition from state $r$ to $s$ in group $k$ ($r, s \in \{0 = \text{non-member}, 1 = \text{member}\}$).

**Sampling link affinities** $\Theta$. Once node group memberships $Z$ are determined, we update the entries of link affinity matrices $\Theta_k$. Direct sampling of $\Theta$ is intractable because of non-conjugacy of the logistic link function. An appropriate method in such case would be the Metropolis-Hastings that accepts or rejects the proposal based on the likelihood ratio. However, to avoid low acceptance rates and quickly move toward the mode of the posterior distribution, we develop a method based on Hybrid Monte Carlo (HMC) sampling [3]. We guide the sampling using the gradient of log-likelihood function with respect to each $\Theta_k$. Because links $Y_{ij}^{(t)}$ are generated independently given group memberships $Z$, the gradient with respect to $\Theta_k[x, y]$ can be computed by

$$-\frac{1}{2\sigma^2}\Theta_k^2 + \sum_{i,j,t} \left( Y_{ij}^{(t)} - p_{ij}^{(t)} \right) \mathbf{1}\{z_{ik}^{(t)} = x, z_{jk}^{(t)} = y\}. \tag{5}$$

**Updating density parameter** $\epsilon$. Parameter vector $\epsilon$ is defined over a finite dimension $T$. Therefore, we can update $\epsilon$ by maximizing the log-likelihood given all the other variables. We compute the gradient update for each $\epsilon_t$ and directly update $\epsilon_t$ via a gradient step.

**Updating hyperparameters.** The number of groups over all time periods is given by a Poisson distribution with parameter $\lambda(1 + \gamma(T - 1))$. Hence, given $\gamma$ we sample $\lambda$ by using a Gamma conjugate prior. Similarly, we can use the Beta conjugate prior for the group death process (*i.e.*, Bernoulli distribution) to sample $\gamma$. However, hyperparameters $\alpha$ and $\beta$ do not have a conjugate prior, so we update them by using a gradient method based on the sampled values of $a_k$ and $b_k$.

**Time complexity of model parameter estimation.** Last, we briefly comment on the time complexity of our model parameter estimation procedure. Each sample $z_{ik}^{(t)}$ requires computation of link probability $p_{ij}^{(t)}$ for all $j \neq i$. Since the expected number of active groups at each time is $\lambda$, this requires $O(\lambda N^2 T)$ computations of $p_{ij}^{(t)}$. By caching the sum of link affinities between every pair of nodes sampling $Z$ as well as $W$ requires $O(\lambda N^2 T)$ time. Sampling $\Theta$ and $\epsilon$ also requires $O(\lambda N^2 T)$ because the gradient of each $p_{ij}^{(t)}$ needs to be computed. Overall, our approach takes $O(\lambda N^2 T)$ to obtain a single sample, while models that are based on the interaction matrix between all groups [4, 5, 11] require $O(K^2 N^2 T)$, where $K$ is the expected number of groups. Furthermore, it has been shown that $O(\log N)$ groups are enough to represent networks [16, 18]. Thus, in practice $K$ (*i.e.*, $\lambda$) is of order $\log N$ and the running time for each sample is $O(N^2 T \log N)$.

## 6 Experiments

We evaluate our model on three different tasks. For quantitative evaluation, we perform missing link prediction as well as future network forecasting and show our model gives favorable performance when compared to current dynamic and static network models. We also analyze the dynamics of groups in a dynamic social network of characters in a movie "*The Lord of the Rings: The Two Towers.*"

**Experimental setup.** For the two prediction experiments, we use the following three datasets. First, the *NIPS co-authorships network* connects two people if they appear on the same publication in the NIPS conference in a given year. Network spans $T$=17 years (1987 to 2003). Following [11] we focus on a subset of 110 most connected people over all time periods. Second, the *DBLP co-authorship network* is obtained from 21 Computer Science conferences from 2000 to 2009 ($T = 10$) [28]. We focus on 209 people by taking 7-core of the aggregated network for the entire time. Third, the *INFOCOM* dataset represents the physical proximity interactions between 78 students at the 2006 INFOCOM conference, recorded by wireless detector remotes given to each attendee [25]. As in [11] we use the processed data that removes inactive time slices to have $T$=50.

To evaluate the predictive performance of our model, we compare it to three baseline models. For a naive baseline model, we regard the relationship between each pair of nodes as the instance of

| Model | NIPS | | | DBLP | | | INFOCOM | | |
|---|---|---|---|---|---|---|---|---|---|
| | TestLL | AUC | F1 | TestLL | AUC | F1 | TestLL | AUC | F1 |
| Naive | -2030 | 0.808 | 0.177 | -12051 | 0.814 | 0.300 | -17821 | 0.677 | 0.252 |
| LFRM | -880 | 0.777 | 0.195 | -3783 | 0.784 | 0.146 | -8689 | 0.946 | 0.703 |
| DRIFT | -758 | 0.866 | 0.296 | -3108 | 0.916 | 0.421 | -6654 | 0.973 | 0.757 |
| DMMG | **−624** | **0.916** | **0.434** | **−2684** | **0.939** | **0.492** | **−6422** | **0.976** | **0.764** |

Table 1: Missing link prediction. We bold the performance of the best scoring method. Our DMMG performs the best in all cases. All improvements are statistically significant at 0.01 significance level.

independent Bernoulli distribution with $Beta(1,1)$ prior. Thus, for a given pair of nodes, the link probability at each time equals to the expected probability from the posterior distribution given network data. Second baseline is LFRM [21], a model of static networks. For missing link prediction, we independently fit LFRM to each snapshot of dynamic networks. For network forecasting task, we fit LFRM to the most recent snapshot of a network. Even though LFRM does not capture time dynamics, we consider this to be a strong baseline model. Finally, for the comparison with dynamic network models, we consider two recent state of the art models. The DRIFT model [4] is based on an infinite factorial HMM and authors kindly shared their implementation. We also consider the LFP model [11] for which we were not able to obtain the implementation, but since we use the same datasets, we compare performance numbers directly with those reported in [11].

To evaluate predictive performance, we use various standard evaluation metrics. First, to assess goodness of inferred probability distributions, we report the log-likelihood of held-out edges. Second, to verify the predictive performance, we compute the area under the ROC curve (AUC). Last, we also report the maximum F1-score (F1) by scanning over all possible precision/recall thresholds.

**Task 1: Predicting missing links.** To generate the datasets for the task of missing link prediction, we randomly hold out 20% of node pairs (*i.e.*, either link or non-link) throughout the entire time period. We then run each model to obtain 400 samples after 800 burn-in samples for each of 10 MCMC chains. Each sample gives a link probability for a given missing entry, so the final link probability of a missing entry is computed by averaging the corresponding link probability over all the samples. This final link probability provides the evaluation metric for a given missing data entry.

Table 1 shows average evaluation metrics for each model and dataset over 10 runs. We also compute the $p$-value on the difference between two best results for each dataset and metric. Overall, our DMMG model significantly outperforms the other models in every metric and dataset. Particularly in terms of F1-score we gain up to 46.6% improvement over the other models.

By comparing the naive model and LFRM, we observe that LFRM performs especially poorly compared to the naive model in two networks with few edges (NIPS and DBLP). Intuitively this makes sense because due to the network sparsity we can obtain more information from the temporal trajectory of each link than from each snapshot of network. However, both DRIFT and DMMG successfully combine the temporal and the network information which results in better predictive performance. Furthermore, we note that DMMG outperforms the other models by a larger margin as networks get sparser. DMMG makes better use of temporal information because it can explicitly model temporally local links through active groups.

Last, we also compare our model to the LFP model. The LFP paper reports AUC ROC score of ∼0.85 for NIPS and ∼0.95 for INFOCOM on the same task of missing link prediction with 20% held-out missing data [11]. Performance of our DMMG on these same networks under the same conditions is 0.916 for NIPS and 0.976 for INFOCOM, which is a strong improvement over LFP.

**Task 2: Future network forecasting.** Here we are given a dynamic network up to time $T_{obs}$ and the goal is to predict the network at the next time $T_{obs} + 1$. We follow the experimental protocol described in [4, 11]: We train the models on first $T_{obs}$ networks, fix the parameters, and then for each model we run MCMC sampling one time step into the future. For each model and network, we obtain 400 samples with 10 different MCMC chains, resulting in 400K network samples. These network samples provide a probability distribution over links at time $T_{obs} + 1$.

Table 2 shows performance averaged over different $T_{obs}$ values ranging from 3 to $T$-1. Overall, DMMG generally exhibits the best performance, but performance results seem to depend on the dataset. DMMG performs the best at 0.001 significance level in terms of AUC and F1 for the NIPS dataset, and at 0.05 level for the INFOCOM dataset. While DMMG improves performance on AUC

| Model | NIPS | | | DBLP | | | INFOCOM | | |
|---|---|---|---|---|---|---|---|---|---|
| | TestLL | AUC | F1 | TestLL | AUC | F1 | TestLL | AUC | F1 |
| Naive | -547 | 0.524 | 0.130 | -3248 | **0.668** | 0.243 | -774 | 0.673 | 0.270 |
| LFRM | -356 | 0.398 | 0.011 | -1680 | 0.492 | 0.024 | -760 | 0.640 | 0.248 |
| DRIFT | **−148** | 0.672 | 0.084 | **−1324** | 0.650 | 0.122 | -661 | 0.782 | 0.381 |
| DMMG | -170 | **0.732** | **0.196** | -1347 | 0.652 | **0.245** | **−625** | **0.804** | **0.392** |

Table 2: Future network forecasting. DMMG performs best on NIPS and INFOCOM while results on DBLP are mixed.

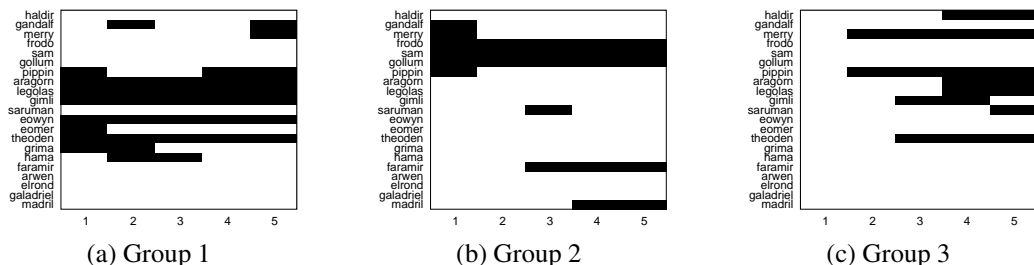

(a) Group 1    (b) Group 2    (c) Group 3

Figure 3: Group arrival and departure dynamics of different characters in the Lord of the Rings. Dark areas in the plots correspond to a give node's (y-axis) membership to each group over time (x-axis)
.

(9%) and F1 (133%), DRIFT achieves the best log-likelihood on the NIPS dataset. In light of our previous observations, we conjecture that this is due to change in network edge density between different snapshots. On the DBLP dataset, DRIFT gives the best log-likelihood, the naive model performs best in terms of AUC, and DMMG is the best on F1 score. However, in all cases of DBLP dataset, the differences are not statistically significant. Overall, DMMG performs the best on NIPS and INFOCOM and provides comparable performance on DBLP.

**Task 3: Case study of "The Lord of the Rings: The Two Towers" social network.** Last, we also investigate groups identified by our model on a dynamic social network of characters in a movie, *The Lord of the Rings: The Two Towers*. Based on the transcript of the movie we created a dynamic social network on 21 characters and $T$=5 time epochs, where we connect a pair of characters if they co-appear inside some time window.

We fit our model to this network and examine the results in Figure 3. Our model identified three dynamic groups, which all nicely correspond to the Lord of the Rings storyline. For example, the core of Group 1 corresponds to Aragorn, elf Legolas, dwarf Gimli, and people in Rohan who in the end all fight against the Orcs. Similarly, Group 2 corresponds to hobbits Sam, Frodo and Gollum on their mission to destroy the ring in Mordor, and are later joined by Faramir and ranger Madril. Interestingly, Group 3 evolving around Merry and Pippin only forms at $t$=2 when they start their journey with Treebeard and later fight against wizard Saruman. While the fight occurs in two separate places we find that some scenes are not distinguishable, so it looks as if Merry and Pippin fought together with Rohan's army against Saruman's army.

**Acknowledgments**

We thank Creighton Heaukulani and Zoubin Ghahramani for sharing data and code. This research has been supported in part by NSF IIS-1016909, CNS-1010921, IIS-1149837, IIS-1159679, IARPA AFRL FA8650-10-C-7058, Okawa Foundation, Docomo, Boeing, Allyes, Volkswagen, Intel, Alfred P. Sloan Fellowship and the Microsoft Faculty Fellowship.

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

## Footnotes

[1]$g(x) = \exp(x)/(1 + \exp(x))$
