[Reviews · NeurIPS 2013]

Submitted by Assigned_Reviewer_2

The paper presents a nonparametric multi-group membership model for dynamic networks. The model captures (1) the birth & death of groups via a distance dependent IBP, (2) the evolution of individual node group membership via a Factorial HMM, and (3) the connectivity structure of groups. MCMC is used for inference.

Runtime for each sample is O(N^2 T log N). This approach is not scalable.

Missing relevant reference:
G. Palla, A.-L. Barabási, T. Vicsek: Quantifying social group evolution. Nature 446:7136, 664-667 (2007).

The results on the future network forecasting are not conclusive. More discussion was needed in the section.

The experimental tasks were made easier by focusing on the several hundred of the most connected people over all time periods.
Summary: The paper is clearly written. It misses some seminal relevant papers from the field of network science like Palla, Barabasi, Vicsek's Nature 2007 paper. The experimental results on network forecasting are mixed.

Submitted by Assigned_Reviewer_5

SUMMARY:
This paper presents a Bayesian nonparametric model for modeling dynamic networks. In particular, the authors present a multi-group membership model based on a distance dependent Indian Buffet Process, where each node in the graph can belong to multiple groups simultaneously (and independently), and the respective group memberships of two nodes determine the probability of an edge existing between them. Unlike previous work, the authors directly model the birth and death dynamics of individual groups, removing ambiguity from the scenario where a group dies and another group is born, such that it is easily differentiated from changes in group membership. The authors present their model, as well as an MCMC inference algorithm for sampling from the posterior. Quantitative results on common, albeit small, data sets (e.g., NIPS, DBLP) show that the proposed model outperforms state-of-the-art alternatives, and a cute example from Lord of the Rings shows qualitatively what the model can do.

The primary strength of this paper is that the authors very clearly describe an interesting model, addressing an important problem, and show that it is better than state-of-the-art alternatives. Crucially, all of this is beautifully written, making the paper a pleasure to read and easy to understand.
The primary weakness of this paper is that the data sets in the experimental section are quite small, although this is par for the course for papers in this field.

CLARITY:
As mentioned above, I found this paper to be beautifully written and very well-presented. There is one key typo, however, on lines 127 and 128. I believe the "\gamma = 1" and "\gamma = 0" should be swapped here, in order for this to make sense.

QUALITY:
As far as I can tell, everything is very well-executed in this paper. The model is well-described, the sampling method is sound, and the experiments are thorough (and include statistical significance checks!). The only problem I have is that the data sets used for the experiments are quite small. I understand that this is usually the case for papers in this domain, but it would be great if there was more discussion on how to scale this model to larger settings.

ORIGINALITY:
The authors do a good job positioning their work in the landscape of similar research, and I find their work to be sufficiently original.

SIGNIFICANCE:
The small data sets in the experiments would probably prevent a practitioner from directly using this approach, but there are enough good ideas in this paper that are empirically shown to improve the state-of-the-art that I can see researchers building upon this work.
Summary: Beautifully written paper on a Bayesian nonparametric model of network dynamics, with experimental results improving upon the state of the art. Data sets are a bit small, however.

Submitted by Assigned_Reviewer_7

Nonparametric Multi-group Membership Model for Dynamic Networks

The paper proposes a new time varying multiple-membership network
model. The multiple group membership is modelled using an IBP and the
time evolution is modelled by a hidden Markov model. In addition,
groups can be born or die according to a stochastic process.

Modeling relational data using Bayesian nonparametrics has recently
received a lot of attention, and this paper makes a small but
significant step towards better modeling the dynamic behaviour of
evolving networks.

In my mind, the proposed model makes perfect sense as a general and
very expressive model of time varying networks. The only thing that I
have some trouble understanding is the argument for having the birth
death process in the model. At every time-step there are an infinite
number of groups (according to the IBP), and a finite number of the
groups have members associated with them. The HMMs tie these IBPs
together, and if all members at some point leave a group it will just
be empty. Later, members can again join this empty group (or join
another empty group). These two situations are disambiguated by the
group parameters (link affinities) so the inference will determine
wheter a new group should be formed or the old group should be
"reused". The exact same is the case for the model with the
birth-death process, only this is a more complicated model. Perhaps I
misunderstand something?

I think the paper is very well written, and the argumentation (except
the above mentioned) was easy to follow. There is a nice balance
between background information, model specification, inference
details, and experimental validation. Some details regarding the
inference is left out (available in an extended version which I did
not examine) and I think this is a reasonable choice, since this is
not essential for understanding the main arguments.

A few additional comments:

- The proposed model, like the LFRM model of Miller et al. [23] is
based on a logistic link function which is the reason that the
computational complexity of inference is at least N^2. I suspect
this prohibits the use of the proposed model to medium or large
scale network analysis. Using e.g. ideas in [24] the computational
complexity could be reduced significantly, which could be an
interesting topic of future research.

- Is there any reason why the density parameter epsilon is optimized
rather than sampled?

- You might be interested in reading Herlau et al. Modeling Temporal
Evolution and Multiscale Structure in Networks, ICML 2013, which is
closely related to your work.

I look forward to hearing the authors' response to my comments.

UPDATE: After reading the authors' comments I stand by my assessment. Just to clarify, I did not mean to say the use of the birth-death process was wrong, only that I find this part of the model unnecessary and not very elegant.
Summary: A very well written, interesting, incrementally novel paper that
combines a multiple-membership relational model with a hidden Markov
model to capture time dependency in networks.
Author Feedback

Author rebuttal: We would like to thank the reviewers for their careful readings and detailed comments. We appreciate the comments, which we will include in the final version of the paper. We believe that reviewers' feedback points to a path for making the paper clearer and increasing its potential impact in this community.

The main reservation shared by the reviewers appears to be related to the scalability of our methodology and the size of the datasets that we used for our experiments.

We absolutely agree with the comments, and our aim was not to claim the algorithm to be scalable but to be very explicit about the limitations of our approach. As Reviewer 5 nicely points out, the scalability of our inference procedure is similar to that of other dynamic network models. In fact, our model is a bit faster than the other models. Our model scales better than the present models, like DRIFT[11] and LFP[13], by a factor of O(log N). We will make sure to more explicitly address this in the final version.

A related comment was about the size of the datasets. As reviewers correctly point out, we tested our algorithms on small (but standard) datasets used by previous approaches (like DRIFT and LFP). The main reason for this is that we wanted to be able to compare our performance to those methods (which used the same datasets). However, we also experimented on larger datasets. For example, we were easily able to run our algorithm on a phone-call network of 4,000 nodes (which is 20 times larger than the largest dataset reported in the paper). Due to space constraints we did not report on these experiments in the original submission. For the final version of the paper we will make sure to also mention the results of these experiments.

Last, Reviewer 7 nicely suggested that the link function from [24] would further reduce the time complexity. In fact, we had thought about the same idea, but we realized that with this link function we loose the flexibility of being able to model the link structure between members and non-members of a given group. However, for networks with strong homophily, we think that the suggested link function [24] would work well. This is a nice direction for future work, which we will definitely discuss in the final version.

Now we address individual reviewer's comments:

* Reviewer 2:
- We thank the reviewer for the reference. We were definitely aware of this work and will consider including the reference in the final version.
- To clarify the network forecasting experiments, when the other models seem to outperform our model, the performance gap is not statistically significant. In contrast, whenever our model outperforms the others the improvement is always statistically significant. Based on this evidence we then made our argument. We will further clarify this in the final paper.

* Reviewer 5:
- We thank the reviewer for pointing out that gamma values were swapped.

* Reviewer 7:
- Unlike previous work, we directly model the birth and death dynamics of individual groups, removing ambiguity from the scenario where a group dies and another group is born, such that these two groups can be easily differentiated by the memberships of each group. Consider an example where we have two disjoint sets of people A and B. At time T group A dies, and at T+1 group B is born. Further assume that connectivity structure of groups A and B is very similar. In this case HMM would “recycle” the group and model A and B as a single group, where between times T and T+1 everyone from A departs the group and everyone from B joins. In contrast, our model would say that group A died and then group B was born. We consider this to be a more accurate representation of reality. For example, social groups are defined based on shared experience. Thus, if there is no shared experience (i.e., overlap in group membership) then we consider that to be a new different group. We can further elaborate on this in the final version of the paper.
- The reason we optimize rather than sample epsilon is for the fast convergence, since we use the gradient method by optimization.
- We thank the reviewer for the ICML reference!